# Investigation of Crystallization and Salt Formation of Poorly Water-Soluble Telmisartan for Enhanced Solubility

**DOI:** 10.3390/pharmaceutics11030102

**Published:** 2019-02-28

**Authors:** Chulhun Park, Nileshkumar M. Meghani, Yongkwan Shin, Euichaul Oh, Jun-Bom Park, Jing-Hao Cui, Qing-Ri Cao, Thao Truong-Dinh Tran, Phuong Ha-Lien Tran, Beom-Jin Lee

**Affiliations:** 1College of Pharmacy, Ajou University, Suwon 16499, Korea; chulhunp1020@gmail.com (C.P.); meghani.nilesh@gmail.com (N.M.M.); luminurs@hanmail.net (Y.S.); 2College of Pharmacy, The Catholic University of Korea, Bucheon 14662, Korea; eoh@catholic.ac.kr; 3College of Pharmacy, Sahmyook University, Seoul 01795, Korea; junji4@gmail.com; 4College of Pharmaceutical Sciences, Soochow University, Suzhou 215006, China; jhcui@suda.edu.cn (J.-H.C.); qrcao@suda.edu.cn (Q.-R.C.); 5Department for Management of Science and Technology Development, Ton Duc Thang University, Ho Chi Minh City, Vietnam; trantruongdinhthao@tdt.edu.vn; 6Faculty of Pharmacy, Ton Duc Thang University, Ho Chi Minh City, Vietnam; 7School of Medicine, Deakin University, Waurn Ponds, VIC 3216, Australia; thlphuong1601@gmail.com

**Keywords:** telmisartan, poorly water-soluble drug, salt formation, solubility enhancement, drug crystallinity, stability

## Abstract

The crystal changes and salt formation of poorly water-soluble telmisartan (TEL) in various solvents were investigated for enhanced solubility, stability and crystallinity. Polymorphic behaviors of TEL were characterized by dispersing in distilled water, acetone, acetonitrile, DMSO, or ethanol using Method I: without heat and then dried under vacuum at room temperature; and Method II: with heat below boiling temperature, cooled at 5 °C, and then dried under vacuum at 40 °C. For salt formation (Method III), the following four powdered mixtures were prepared by dispersing in solution of hydrochloric acid (HCl) (pH 1.2), TEL/HCl; in simulated gastric fluid (pH 1.2 buffer), TEL/simulated gastric fluid (SGF); in intestinal fluid (pH 6.8 buffer), TEL/simulated intestinal fluid (SIF); or in NaOH (pH 6.8), TEL/NaOH, respectively, and then dried under a vacuum at room temperature. The structures of powdered mixtures were then studied using a field emission scanning electron microscope (FESEM), differential scanning calorimetry (DSC), powder X-ray diffraction (PXRD), FTIR, ^1^H nuclear magnetic resonance (^1^H-NMR), and LC–MS. The solubility of TEL in powdered forms was performed in pH 6.8, pH 1.2, and distilled water. No polymorphic behaviors of TEL were observed in various solvents as characterized by FESEM, DSC, PXRD, and FTIR. However, the structural changes of powdered mixtures obtained from Method III were observed due to the formation of salt form. Moreover, the solubility of salt form (TEL/HCl) was highly increased as compared with pure TEL. There were no significant changes of TEL/HCl compared with TEL in the content assay, PXRD, DSC, and FTIR during stressed storage conditions at 40 °C/75% relative humidity (RH) for 4 weeks under the closed package condition. Therefore, the present study suggests the new approach for the enhanced stability and solubility of a poorly water-soluble drug via salt form.

## 1. Introduction

Polymorphism is one of the popular ways for improving physicochemical properties of various drugs. In general, polymorphs have different physicochemical properties, such as melting point, solubility, and density [1]. For example, the differences of physical properties of mefenamic acid polymorphs in various solvents may affect the reproducibility of the manufacturing process of dosage forms and their performance [2]. 

Salt formation is a commonly used method for enhancing physicochemical properties of various drugs, including solubility [3]. Among commonly used salts include sodium, hydrochloride, phosphate, salicylate, and so on for active pharmaceutical ingredient (API) salt selection in pharmaceutical industries [4]. Salt forms are mainly affected by physiochemical properties of API [5]. There are many anions and cations that can be used to form salts with weakly basic or acidic drugs. For example, hydrochloride is by far the most frequently used to form the salt of cationic drug. Salt formation is similar with co-crystal, which is being mediated by proton transfer from one partner molecule to the other [6,7,8]. Stability of the polymorph and salt formation is also known to be an important issue because the storage conditions, such as temperature, humidity, or pharmaceutical excipients, affect the stability of crystal forms [8,9,10]. In particular, stability issues of APIs have been addressed by formulating with pH modifiers or considering the changes of physicochemical properties [11]. 

Telmisartan (TEL) was chosen as a model drug, which is labeled as an angiotensin II-receptor antagonist (ARA-II) and selectively blocks the receptor of angiotensin II [12]. The ARA-IIs are safe and effective agents in the treatment of hypertension and heart failure. They selectively block the AT1 angiotensin II receptor, are long-acting, and have a good tolerability profile [13]. Additionally, the pharmacokinetic behaviors and clinical properties of TEL have reported that TEL can be a partial agonist of Peroxisome proliferator-activated receptor γ (PPARγ), and it can simultaneously block the angiotensin II receptor and activate PPARγ to effectively treat diabetes and cardiovascular disease [14]. Physiochemical properties of TEL were summarized in Table 1 [15]. 

Due to its high lipophilicity and a high volume distribution, TEL shows a good tissue penetration [16]. However, the solubility of TEL is dependent on pH of solution. The solubility of an API can significantly affect the drug absorption and, therefore, its bioavailability [17,18]. Various approaches for solubilization and stabilization of TEL have been formulated by modulating microenvironmental pH of compositions [19,20]. However, modification of drug structures via polymorphism and/or salt formation is challenging to improve physicochemical properties of poorly water-soluble drugs due to the complex process associated with stability and purity of TEL [21]. TEL polymorphic forms in the presence of corrosive agents, such as formic acid and acetic acid, exhibited unique physical properties when examined via differential scanning calorimetry (DSC), powder X-ray diffraction (PXRD), and FTIR [22]. However, the chemical structure of TEL was quickly degraded within a short period of time, resulting in less efficacy and increased toxic impurities. To date, no detailed investigation about mechanism of TEL salts formation has been reported based on structural changes. The commercial product Micardis^®^ and various researches of TEL were associated with salt formation of sodium or alkalizing agent using solid dispersion technology [19,23]. However, these studies still contain limited knowledge regarding crystallization and salt formation of TEL, as well as effective control of hygroscopicity without Alu/Alu packaging. 

The aim of this study was to investigate structural changes and salt formation of TEL in various solvent media and characterize their physicochemical properties. The screening studies for a new crystalline form of TEL were conducted to evaluate polymorphic changes. TEL salt forms with different aqueous solvent were obtained through various processes (reflux, distillation, filtration, and concentration). The structural changes of TEL salt forms were then characterized by various instrumental analyses using FESEM, DSC, PXRD, FTIR, ^1^H-NMR, and LC–MS. Furthermore, solubility and stability of the TEL salt form (TEL/hydrochloric acid (HCl)) and its mechanistic understanding were evaluated.

## 2. Experimental

### 2.1. Materials

TEL was purchased from NJMMM Co., Ltd. (Nanjing, China). Acetone and DMSO were purchased from Showa (Tokyo, Japan). HPLC-grade methanol and acetonitrile were obtained from Fisher Scientific Korea Ltd. (Seoul, Korea). First-grade ethanol was supplied from Duksan Reagent & Chemicals (Gyeonggi-do, Korea). All other chemicals were of reagent grade and used without further purification.

### 2.2. Methods

#### 2.2.1. Preparation of Powdered Mixtures of TEL

##### Preparation without Heat (Method I)

One gram of TEL was dispersed in 50 mL acetone, acetonitrile, DMSO, ethanol, distilled water, pH 1.2 buffer, or pH 6.8 phosphate buffer, respectively. After dispersing the solution in each solvent for 1 hr using a magnetic stirrer, 50 mg of activated charcoal was added in the solution. The resulting mixture was filtered by 10 µm filter paper. The residue was washed three times with each solvent. After stirring for 24 hr at about room temperature (~23 °C), the resulting solution was subsequently dried under a vacuum at room temperature for 24 hr. The dried powders were passed through 60-mesh sieve.

##### Preparation with Heat (Method II)

One gram of TEL was dispersed in 50 mL acetone, acetonitrile, DMSO, ethanol or distilled water, pH 1.2 buffer or pH 6.8 phosphate buffer, respectively. Each mixture solutions were heated for 1 hr and sufficiently stirred using thermally controlled magnetic stirrer (acetone −40 °C, acetonitrile, ethanol −70 °C, DMSO, distilled water, pH 1.2 buffer, pH 6.8 phosphate buffer −90 °C). 50 mg of activated charcoal was added in the solution. The resulting mixture was filtered by 10 µm filter paper. The residue was washed three times with each solvent. The resulting solution was then cooled at 5 °C and final resulting mixtures dried under vacuum at 40 °C for 24 hr. The dried powders were passed through 60-mesh sieve. 

##### Salt Formation (Method III)

One gram of TEL was suspended in 50 mL of each solvent (simulated gastric fluid (SGF), simulated intestinal fluid (SIF), 0.1 M HCl solution, and 0.1 M sodium hydroxide solution, respectively). SGF (pH 1.2 ± 0.05) consists of 0.2% (w/v) sodium chloride and 0.7%(v/v) concentrated HCl; while, SIF (pH 6.8 ± 0.05) was prepared with 340.25 mg potassium dihydrogen phosphate and 44.8 mg sodium hydroxide in 50 mL. Each mixture was refluxed and about 50 mL of solvent was distilled off. The remaining residue was gradually combined with 15 mL of water at 40 °C. After stirring for 1 hr with a magnetic stirrer at about 23 °C, the remaining residue was slowly combined with 50 mg of activated charcoal. The resulting mixture was filtered by 10 µm filter paper. After filtration, the transparent TEL salt form solution was obtained, and 2.5 mL of acetone was added to solution. Then, the final solution was dried under a vacuum at 40 ℃ for 24 hr. The dried powders were passed through 60-mesh sieve. 

#### 2.2.2. Solubility Determination

Solubility of TEL and its powdered mixtures were investigated in various solutions. An excess amount of TEL and its powdered mixtures were added into distilled water, simulated gastric fluid (pH 1.2), and simulated intestinal fluid (pH 6.8), respectively, in a microtube. The mixture was placed at 37 °C for 48 hr. An aliquot was centrifuged at 10,000 rpm for 10 min. The content of TEL in supernatants, defined as the degree of solubility in this study, was assayed by HPLC from a standard calibration curve.

#### 2.2.3. Stability Study

For the conventional stability studies, powdered samples were stored for up 4 weeks under the following stressed condition: 40 °C/75% relative humidity (RH). Samples were then measured by HPLC, DSC, PXRD, and FTIR method.

#### 2.2.4. DSC Study

The thermal behaviors of TEL and its powdered mixtures were investigated using DSC 2910 (TA Instruments, Dupont, Hayward, CA, USA). About 5 mg of resulted product was weighed in a standard open aluminum pan. An empty pan of the same type was utilized as a reference. The resulting product was heated from 25 to 300 °C at a heating rate of 10 °C/ min with nitrogen as purge gas. The flow rate of purging gas with nitrogen was 50 mL/min. The calibration of temperature and heat flow was performed within indium [24,25].

#### 2.2.5. PXRD Study

PXRD patterns of TEL and its powdered mixtures were obtained using a D5005 (Bruker, Karlsruhe, Germany) with Cu-K radiation at 40 KV 50 mA by scanning in steps of 0.02° from 3° to 40° with a rate of one second per step, using a sample holder.

#### 2.2.6. FTIR Study

Potassium bromide (KBr) pellets were prepared by gently mixing 1 mg of TEL and its powdered mixtures with 200 mg KBr. The FTIR spectrum (400 ~ 4000 cm^−1^) was obtained with a resolution of 2 cm^−1^ using an Excalibur series FTS 3000 (Bio-Rad, Cambridge, UK).

#### 2.2.7. SEM Analysis

The morphologies of TEL and its powdered mixtures were observed using an SEM (JSM-5410, JEOL, Tokyo, Japan) at an acceleration voltage of 15 kV during scanning. The samples were coated with a thin layer of gold for 10 min.

#### 2.2.8. ^1^H-NMR Analysis

All ^1^H-NMR spectra were obtained using a Bruker AMX-600 instrument (Bruker, Karlsruhe, Germany). The solution of TEL and its powdered mixtures (50 mM) in methanol-d4 was measured at 298 K. The instrument was set to acquire at least 128 scans for a proton spectrum. The spectrum was processed using a line broadening of 0.3 Hz. Phasing and baseline corrections were applied.

#### 2.2.9. LC–MS Analysis

LC–MS was applied to identify chemical structures of TEL and its powdered mixtures. Their mass spectra were acquired using a Finnigan TSQ Quantum Ultra AM triple stage quadrupole mass spectrometer (Thermo Electron Corporation, San Jose, CA, USA) equipped with electrospray ionization (ESI) interface. The vaporizer temperature was set to 100 °C and nitrogen was applied as the sheath gas. The heated capillary was maintained at 350 °C. Mass analysis was performed in the positive ion mode with the source current set at 5 mA, and the potential of tube lens was set at 82 volts. The m/z scanning ranged from 100 to 700. The type of column and composition of mobile phase were consistent with the conditions described above. Mass parameters were tuned in both positive and negative ionization modes for the analytes. Clear relative absorbance of TEL and the TEL/HCl sample solution was achieved in positive ionization mode. Data from the multiple react ions monitoring were considered to obtain better selectivity. The protonated form of each analyte ion was the parent ion in the Q 1 spectrum and was used as the precursor ion to obtain Q 3 product ion spectra. Based on the literature Reference [26], 40.00 mg of TEL (MW = 514.629g/mol) and 42.83mg of TEL/HCl (MW = 551.087 g/mol) were accurately weighed on a 100 mL volumetric flask. Each sample was dissolved in the mobile phase and adjusted at a concentration of 4.00 ng/mL as TEL. 

#### 2.2.10. HPLC Analysis of TEL and Its Related Products

A reverse phase HPLC system was used for TEL analysis. The HPLC system (Waters Corp., Milford, MA, USA) consisted of a pump (Waters™ 600 Controller), an autosampler (Waters™ 717 plus Autosampler), a degasser (Waters™ In-line Degasser), and an analytical column (Gemini 5 μ, C18, 4.6 × 150 mm) set at 296 nm. The mobile phase used was an ammonium acetate buffer (pH 7.0) and methanol at a ratio of 25:75, and its flow rate was maintained at 1 mL/min. The HPLC data processing was performed using Borwin^®^ software (Version 1.20, Jasco, Groß-Umstadt, Germany).

## 3. Results and Discussion

### 3.1. Characterization of Powdered Mixtures by Method I and II (Polymorphic Behaviors)

Figure 1 and Figure 2 show the DSC thermogram and PXRD patterns of TEL and its powdered mixtures prepared by Method I and II, respectively. The onset temperature of each sample was close to 269 °C without significant change, and the delta enthalpy also showed no significant change. The sharp endothermic peak of powdered mixtures was identical with the intrinsic peaks of TEL. The observed results of PXRD of powdered mixtures was similar to that of the TEL. These results indicated that the crystallinity and polymorphism of TEL was not affected by the types of solvents and preparation method with or without presence of heat. 

Figure 3 represents the solubility profile of TEL and its powdered mixtures in various media. The organic solvents except DMSO did not affect the solubility of TEL significantly, suggesting that no polymorphism nor salt formation of TEL occurred. Since, the samples were formed in buffered solution, the pH was maintained in pH 1.2 and pH 6.8 buffer. Interestingly, the mixtures prepared in a solution of pH 1.2 buffer by both Method I,II showed enhanced solubility via salt formation in three testing media. Moreover, the solubility of TEL showed the highest in SGF (pH 1.2) because of its intrinsic solubility. The solubility of Method I DMSO TEL forms was ranked second after the pH 1.2 buffer solution, mainly in distilled water. Relative polarity of different organic solvents could critically contribute the TEL solubility. It has been reported that TEL solubility in DMSO was significantly higher than many other solvents. Considering the previous studies about solubility and stability data of TEL, the DMSO showed high performance of maintaining stability and dissolving large quantities of TEL. That is why the Method II DMSO TEL had lower solubility in distilled water, because the heat was applied to reduce DMSO polarity.

### 3.2. Characterization of Powdered Mixtures by Method III (Salt Formation)

#### 3.2.1. Instrumental Analysis

Figure 4 shows the DSC thermograms of powdered mixtures prepared by Method III. TEL melting peak was found to be at 269.795 °C; while, onset temperature of TEL/SGF (pH 1.2) and TEL/HCl (pH 1.2) had shifted to 261.061 °C. At pH 1.2, the onset temperature decreased as the degree of entropy increased due to the salt formation between TEL and hydrochloride. In case of TEL/SGF, it was confirmed that another crystal polymorphism of TEL occurred due to the interaction with sodium salt rather than hydrochloride formation as sodium chloride was contained in the SGF solution. However, the onset temperature peak of TEL obtained from dispersed TEL/NaOH (pH 6.8) and TEL/SIF (pH 6.8) was consistent. 

Figure 5 represents the PXRD patterns of TEL and powdered mixtures. Since all samples prepared by Method III showed numerous intrinsic crystalline peaks, no distinct amorphousness was observed. The PXRD peaks of TEL dispersed in TEL/SGF (pH 1.2) and TEL/HCl (pH 1.2) were reduced, as compared with TEL or TEL/SIF and TEL/NaOH. TEL/SIF (pH 6.8) and TEL/NaOH (pH 6.8) had similar drug crystallinity with TEL alone, suggesting no amorphous nature of the formulation. Interestingly, the characteristic sharp X-ray peaks at 30° ~ 35° 2θ of TEL/SIF and TEL/SGF were observed. It was presumed to maintain a sT crystalline form of TEL, but due to the newly formed salt form in SGF and SIF, strong peaks appeared among diffraction patterns. Additionally, extra peaks of TEL/SGF and TEL/SIF appeared due to superior crystallinity of TEL salt forms as compared with TEL/HCl and TEL/NaOH.

Figure 6 shows the FTIR peaks of TEL and powdered mixtures obtained by Method III. FTIR spectra of TEL in TEL/SIF (pH 6.8) and TEL/NaOH (pH 6.8) were not changed with respect to TEL, but the TEL peaks in TEL/SGF (pH 1.2) and TEL/HCl (pH 1.2) were shifted. The C = O stretching band of TEL and TEL/HCl (pH 1.2) was observed at 1693.2 cm^−1^, while the O-H band was observed at 3061.1 cm^−1^ in the case of TEL/HCl (pH 1.2). These two characteristic peaks were indicated by the arrows. Interestingly, the DSC thermograms and PXRD patterns of the powdered mixtures obtained from TEL/SGF (pH 1.2) and TEL/HCl (pH 1.2) were almost identical, while those of TEL/SIF (pH 6.8) and TEL/NaOH (pH 6.8) were unchanged. The presence of HCl in TEL/SGF (pH 1.2) and TEL/HCl (pH 1.2) was a critical factor for salt formation of TEL.

#### 3.2.2. Surface Morphologies of Powdered Mixtures by FESEM

Surface morphologies of TEL and powdered mixtures by Method III are shown in Figure 7. The appearance of TEL powder was found to be rod-shaped using by FESEM. The powdered mixtures from TEL/SGF (pH 1.2) were salt forms with plate-shaped (1–10 µm) while TEL/HCl (pH 1.2) appeared to be in cubic form, providing solubility enhancement (see Figure 8). TEL/NaOH (pH 6.8) was similar in shape to TEL/SIF (pH 6.8). TEL/SIF (pH 6.8) was partially broken into a rod-type form.

#### 3.2.3. Solubility Study

Among the four salt forms, TEL/HCL was selected for further solubility studies. Figure 8 shows the solubility profile of powdered mixtures obtained from TEL/HCl (pH 1.2) in the various media, such as distilled water, SGF (pH 1.2), and SIF (pH 6.8). The powdered mixtures obtained from TEL/HCl solution showed higher solubility in all media. The solubility of powdered mixtures obtained from TEL/HCl was 1243.17 μg/mL in distilled water, much higher compared with TEL (0.09 μg/mL). In SIF (pH 1.2), the solubility of powdered mixtures obtained from TEL/HCl was also 10 times higher (1404.46 μg/mL) than that of TEL (125.41 μg/mL). The enhanced solubility of powdered mixtures obtained from TEL/HCl solution was due to the salt formation of TEL. In SIF (pH 6.8), the solubility of powdered mixtures from TEL/HCl (pH 1.2) was 86.92 μg/mL, while TEL was 0.05 μg/mL. This data indicated the successful salt formation of the powdered mixture of TEL/HCL, which resulted in enhanced solubility of TEL due to its stable crystalline salt formation.

#### 3.2.4. Mechanistic Characterization of Salt Form (TEL/HCl)

##### ^1^H-NMR Analysis

Figure 9 shows the ^1^H-NMR peaks of TEL and powdered mixtures obtained from TEL/HCl solution (pH 1.2). The shift of NMR signals has been affected by slight differences in the sample environments. The NMR peaks of powdered mixtures obtained from the TEL/HCl solution (pH 1.2) were slightly shifted via salt formation of TEL. The ^1^H-NMR data of the TEL and the powdered mixtures obtained from the TEL/HCl solution (pH 1.2) is listed in Table 2. The δ signal of 7.17~7.19 (d, 2H) disappeared, and 7.90~7.94 (d, 1H), 8.06 (s, 1H) appeared in powdered mixtures obtained from the TEL/HCl solution (pH 1.2). The changes of these δ signals indicated molecular interaction of the ionized carboxyl group and the N–H bond between HCl and the part of methylbenzimidazole of TEL [27]. 

##### LC–MS Analysis

LC–MS spectra from a positive ion mode of TEL and powdered mixtures obtained from TEL/HCl solution (pH 1.2) are shown in Figure 10. The ion spectra were measured at the same TEL concentration of 4.0ng/mL, and the relative absorbance was similar with two samples. TEL/HCl salt form was rapidly ionized and the maximum m/z 515 was observed on the positive mode. In fact, ion spectrums of TEL and powdered mixtures obtained from TEL/HCl solution (pH 1.2) having a protonated molecular ion [M + H]^+^ was identified at m/z 515. However, the ion spectrum observed at m/z 167 suggested that the hydrogen bond was formed between methylbenzoimidazole and HCl because the ion spectrum observed at m/z 391 indicated that two hydrogen bonds were formed between methyl-6-(1-methylbenzoimidazol-2-yl)-2-propyl-benzoimidazol and HCl. 

#### 3.2.5. Stability Study

In designing dosage forms, it is essential to investigate the stability of salt forms to ensure pharmaceutical quality and bioavailability [2]. Table 3 shows the percentage of TEL content during stressed storage condition at 40 °C/75% RH for 4 weeks. The content of powdered mixtures obtained from TEL/HCl solution (pH 1.2) was significantly decreased (about 9%) in an open condition for 4 weeks while the content of TEL alone was not significantly changed. The salt form of TEL in powdered mixtures gradually experienced the hydrolysis and oxidation. The hydrolysis is frequently catalyzed by hydrogen ions in the powder of weak acid and salt form [28]. However, the drug content was almost unchanged when the salt form was stored under the closed package condition.

Figure 11 shows the changes of DSC, PXRD and, peaks of TEL and powdered mixtures obtained from TEL/HCl solution (pH 1.2) after stressed storage condition for 4 weeks, respectively. Although drug content of powdered mixtures obtained from TEL/HCl solution decreased, the structural behaviors were consistently maintained during storage for 4 weeks.

## 4. Conclusions

No polymorphic behaviors of TEL were observed in acetone, acetonitrile, ethanol, DMSO, nor distilled water by Method I and II. Contrarily, powdered mixtures obtained from the TEL/HCl solution suggested the formation of salt form as confirmed by the structural changes by DSC, PXRD, FTIR, ^1^H-NMR, and LC–MS solubility studies. The salt form had much higher solubility than that of the TEL. The structures of salt form were found to be stable and consistently maintained during stressed storage conditions at 40 °C/75% RH for 4 weeks. However, the drug content of TEL/HCl salt form was gradually decreased under the open package condition but was found to be unchanged when the salt form was stored under the closed package condition. It was evident that the salt formation of TEL could provide a way to enhance solubility of poorly water-soluble TEL without losing drug crystallinity.

## Figures and Tables

**Figure 1 pharmaceutics-11-00102-f001:**
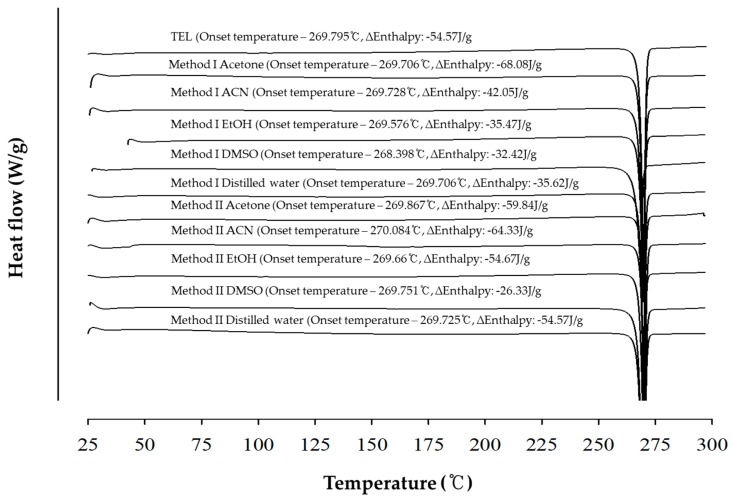
Comparison of the differential scanning calorimetry (DSC) thermograms of TEL and its powdered mixtures prepared by Method I and II.

**Figure 2 pharmaceutics-11-00102-f002:**
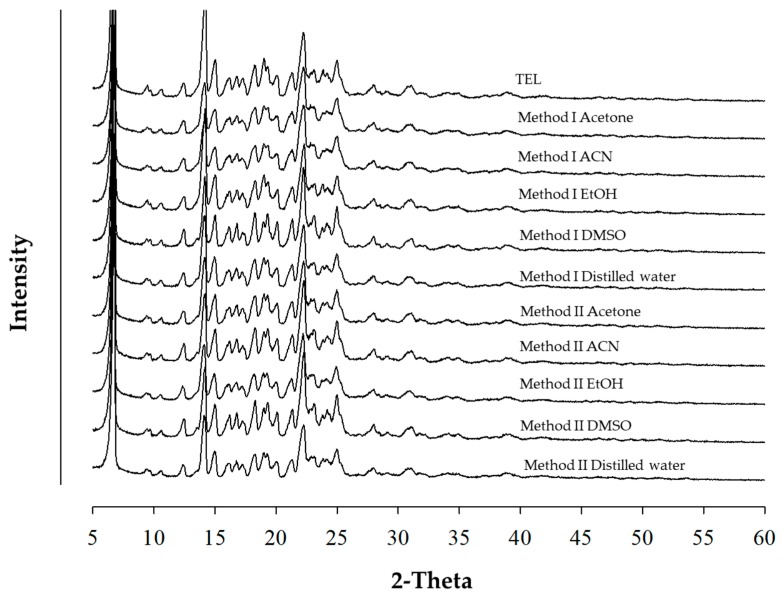
Comparison of powder X-ray diffraction (PXRD) patterns of TEL and its powdered mixtures prepared by Method I and II.

**Figure 3 pharmaceutics-11-00102-f003:**
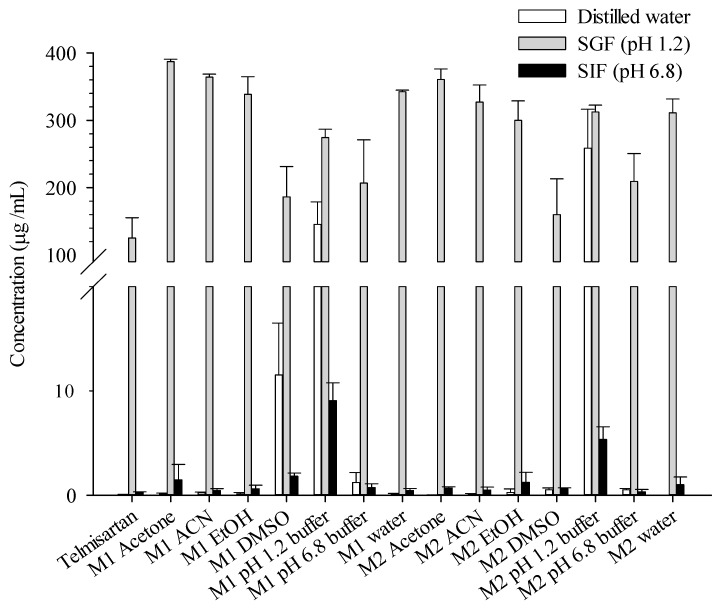
Solubility of TEL and its powdered mixtures prepared in various conditions in distilled water, SGF (pH 1.2), and SIF (pH 6.8) at 37 °C. M1 = Method I, M2 Method II.

**Figure 4 pharmaceutics-11-00102-f004:**
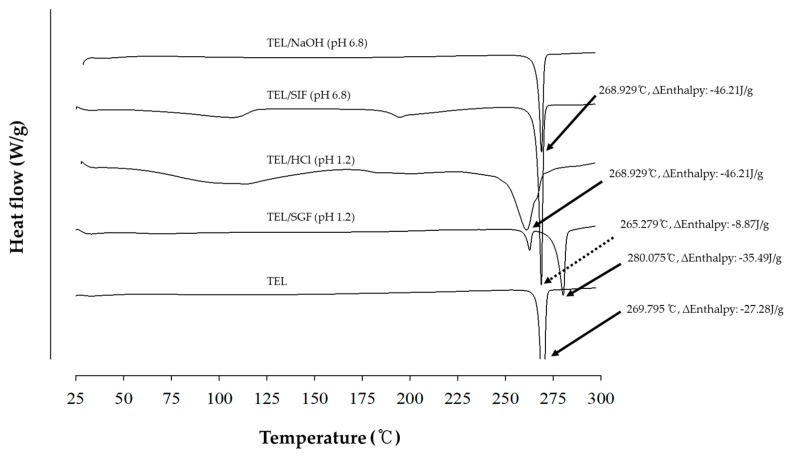
Comparison of the DSC thermograms of TEL and four powdered mixtures obtained from SGF (pH 1.2), SIF (pH 6.8), hydrochloric acid (HCl) (pH 1.2), and NaOH (pH 6.8) prepared by Method III.

**Figure 5 pharmaceutics-11-00102-f005:**
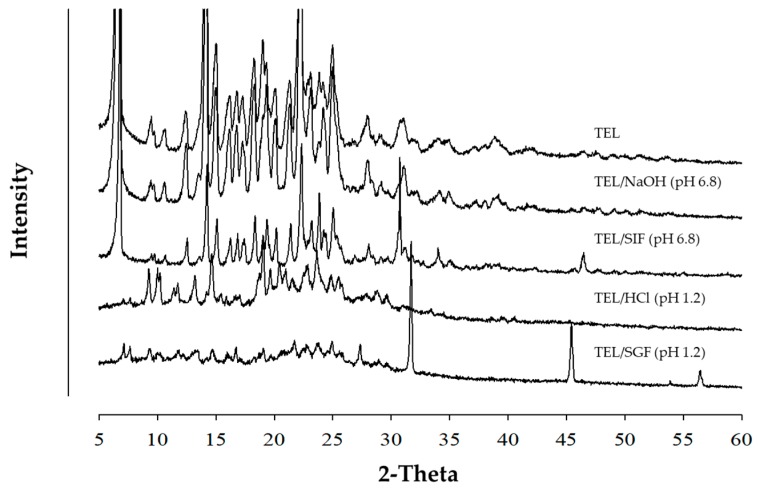
Comparison of PXRD patterns of TEL and four powdered mixtures obtained from simulated gastric fluid (SGF) (pH 1.2), simulated intestinal fluid (SIF) (pH 6.8), HCl (pH 1.2), and NaOH (pH 6.8) prepared by Method III.

**Figure 6 pharmaceutics-11-00102-f006:**
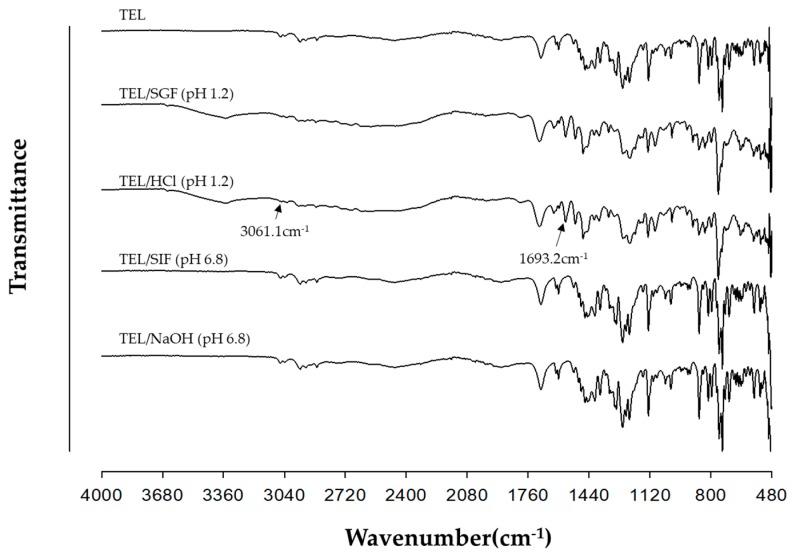
FTIR spectra of TEL and four powdered mixtures obtained from TEL/SGF (pH 1.2), TEL/HCl (pH 1.2), TEL/SIF (pH 6.8), and TEL/NaOH (pH 6.8) prepared by Method III. Two characteristic peaks were indicated by the arrows.

**Figure 7 pharmaceutics-11-00102-f007:**
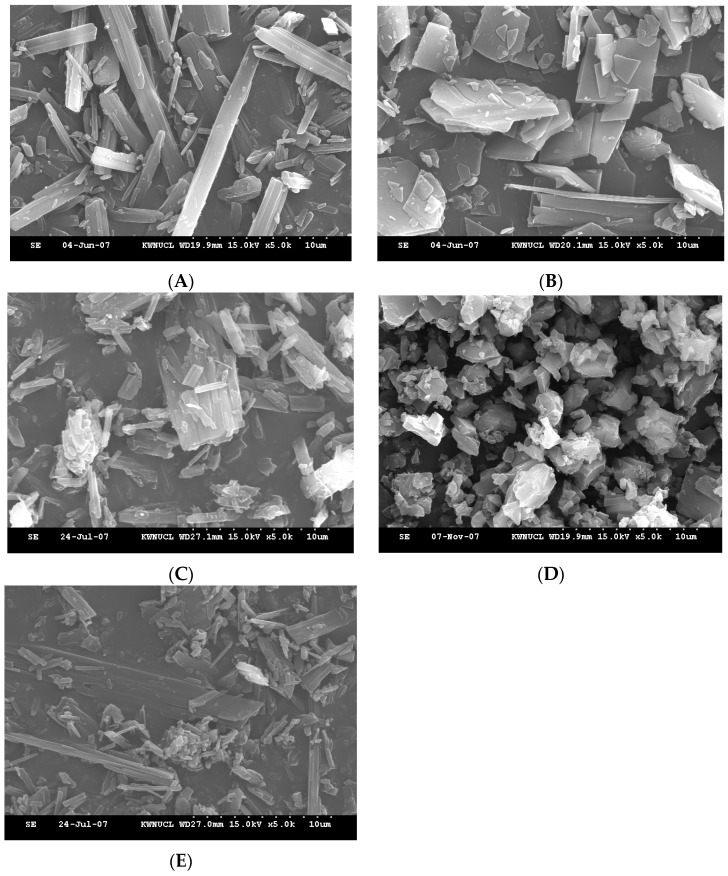
Surface morphologies using SEM of (**A**) TEL and four powdered mixtures obtained from (**B**) TEL/SGF (pH 1.2), (**C**) TEL/SIF (pH 6.8), (**D**) TEL/HCl (pH 1.2), and (**E**) TEL/NaOH (pH 6.8) prepared by Method III (Scale bar: 10µm).

**Figure 8 pharmaceutics-11-00102-f008:**
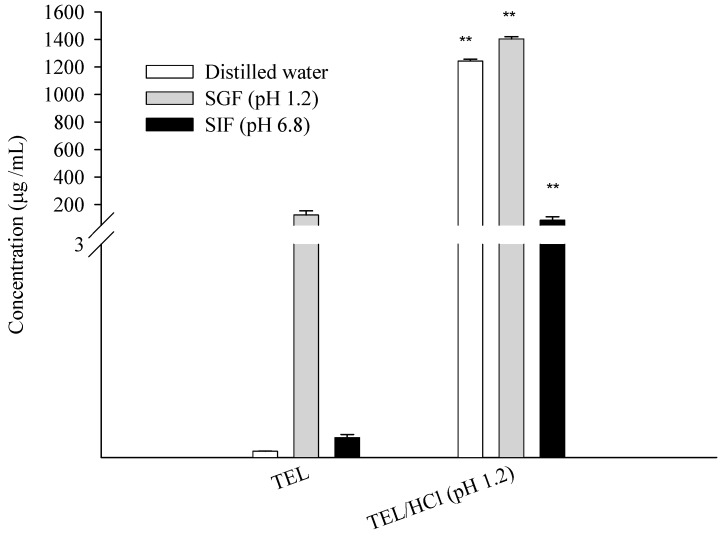
Solubility enhancement of TEL and powdered mixtures obtained from TEL/HCl (pH 1.2) in distilled water, SGF (pH 1.2), and SIF (pH 6.8) at 37 °C. ** Significantly different from TEL (*p* < 0.001).

**Figure 9 pharmaceutics-11-00102-f009:**
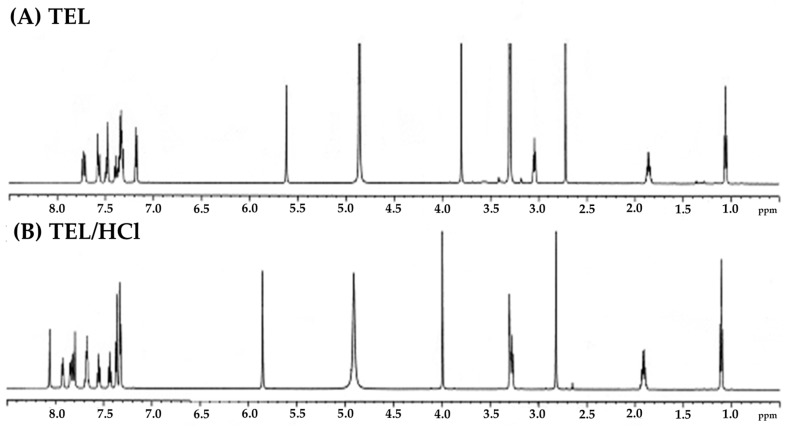
^1^H-NMR spectra of (**A**) TEL and (**B**) TEL/HCl (pH 1.2).

**Figure 10 pharmaceutics-11-00102-f010:**
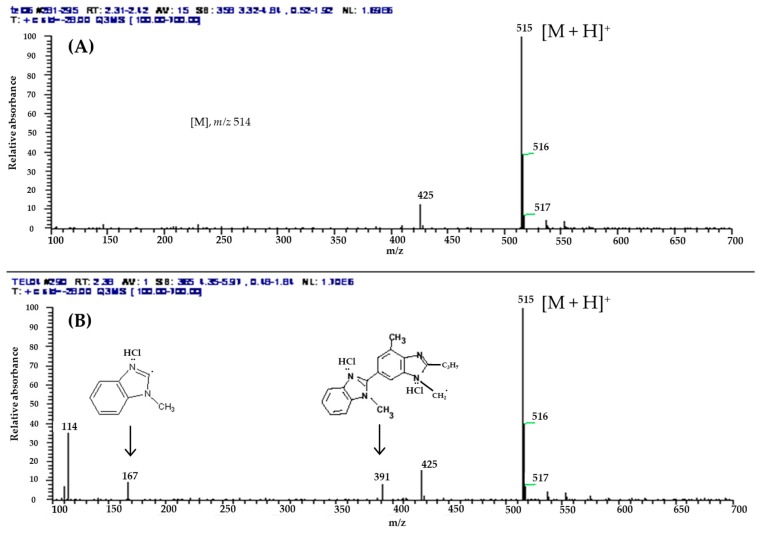
Full-scan product ion LC–MS spectra of [M + H]^+^ for (**A**) TEL and (**B**) TEL/HCl (pH 1.2).

**Figure 11 pharmaceutics-11-00102-f011:**
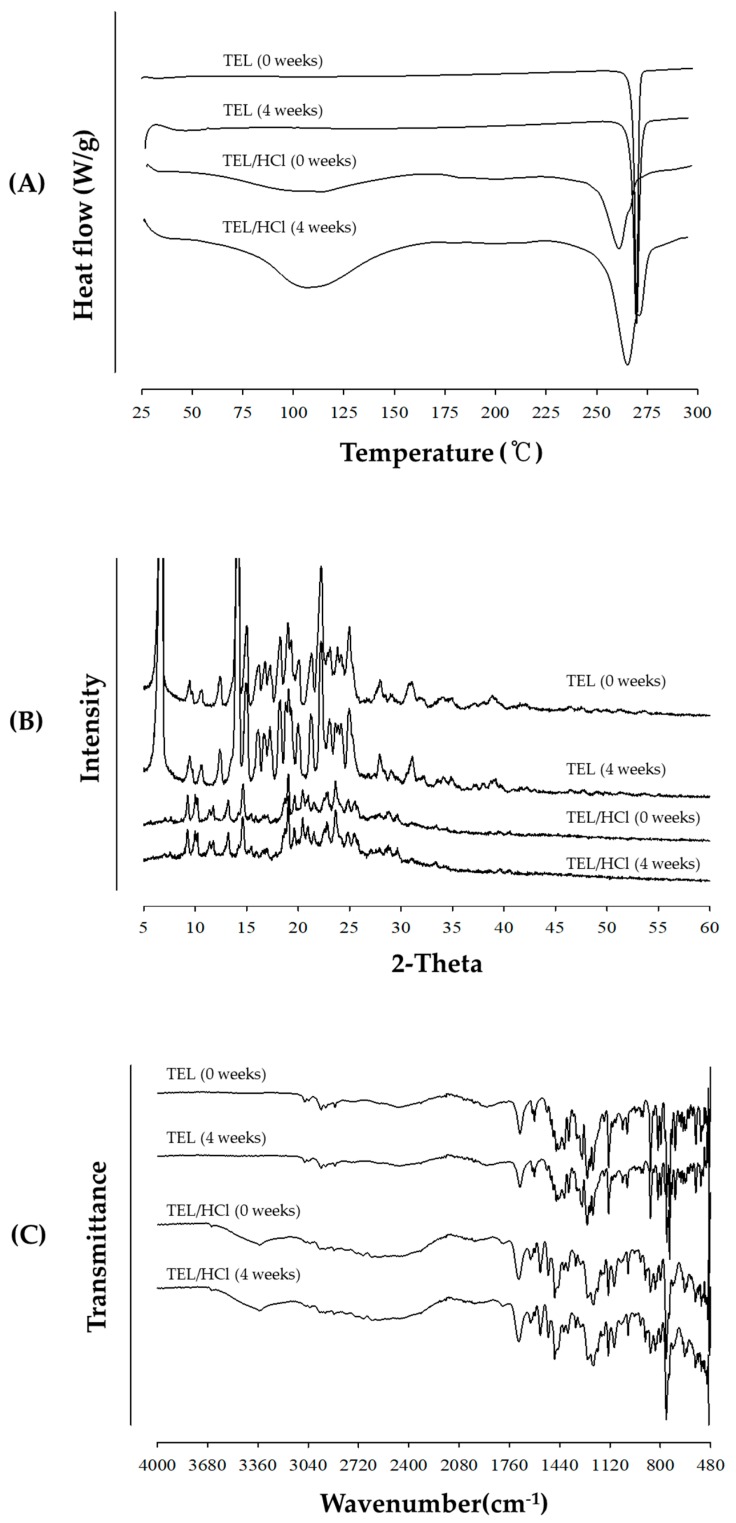
Comparison of (**A**) DSC thermograms, (**B**) PXRD patterns, and (**C**) FTIR spectra of TEL and powdered mixtures obtained from TEL/HCl solution (pH 1.2) during storage at 40 °C/75% RH for 4 weeks.

**Table 1 pharmaceutics-11-00102-t001:** Physicochemical properties of telmisartan (TEL).

Parameters	Telmisartan
Chemical structure	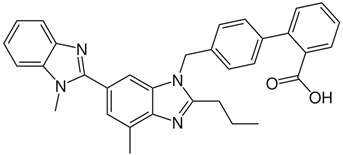
Formula	C_33_H_30_N_4_O_2_
Solubility	Practically insoluble in water high solubility at high/low pH and poor solubility in pH 3–9
Log P	3.2
pKa	3.5, 4.1, 6.0 (weakly acidic)

**Table 2 pharmaceutics-11-00102-t002:** ^1^H-NMR spectral data of the analogues of TEL and TEL/HCl (pH 1.2) (s = singlet, d = duplet, dd = double duplet, t = triplet, si = superimposed).

Sample	^1^H-NMR (methanol-d_4_) δ
TEL	1.05 (t, 3H), 1.79~1.86 (t, 3H), 2.63 (s, 3H), 2.93 (t, 2H), 3.30 (s, 2H), 3.82 (s, 3H), 4.85 (s, 2H), 5.63 (s, 2H), 7.17~7.19 (d, 2H), 7.30~7.37 (dd, 5H), 7.37~7.41 (t, 1H), 7.48 (s, 1H), 7.56 (td, 1H), 7.57~7.59 (d, 1H), 7.70~7.75 (t, 3H)
TEL/HCl (pH 1.2)	1.10 (t, 3H), 1.79~1.86 (t, 3H), 2.81 (s, 3H), 3.20~3.33 (t, 2H), 4.0 (s, 3H), 4.9 (s, 2H), 5.87 (s, 2H), 7.30~7.35 (d, 2H), 7.35~7.38 (d, 2H), 7.41~7.45 (t, 1H), 7.53~7.57 (t, 1H), 7.64~7.70 (td, 1H), 7.77~7.86 (td, 3H), 7.90~7.94 (d, 1H), 8.06 (s, 1H)

**Table 3 pharmaceutics-11-00102-t003:** Assay of TEL and powdered mixtures obtained from TEL/HCl solution (pH 1.2) during storage at 40 °C/75% relative humidity (RH) for 4 weeks. (n = 3)

Types of Sample	% Assay (Mean ± SD)		
	Day 1	Week 4	Package condition
**TEL**	100.00 ± 1.41	98.99 ± 2.30	Open
**TEL/HCl (pH 1.2)**	99.95 ± 0.41	90.90 ± 2.60	Open
**TEL/HCl (pH 1.2)**	99.99 ± 0.37	99.90 ± 2.23	Closed

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
