# Peer review of "Investigation of Crystallization and Salt Formation of Poorly Water-Soluble Telmisartan for Enhanced Solubility"

_pharmaceutics, 2019, doi:10.3390/pharmaceutics11030102_

Round 1
Reviewer 1 Report
The authors have corrected the manuscript accordingly according to the reviewer's comments. But, there are still some necessities to correct.
Method:
In my previous comment,
"How is the composition ratio of drug and acid? If the salt is formed by method III, composition ratio of drug and salt should be calculated. And determine by analytical technique."
There is still no information in the manuscript.
Author answered that the composition was 1:1, but not mentioned in the method.
And my question is not for preparation, what is the actual ratio of drug and HCl in "salt form"?
Also, the detail for preparation method of method I-III should be described.
As it is, no one can reproduce.
MS analysis:
Me previous comment
"The purpose of MS analysis is uncertain. Why the fragmentation was different between TEL and salt form? Just pH condition of the sample solution? Please check same pH condition. Under solution state, if the salt dissolved completely, MS spectrum of TEL salt can be corresponding
to that of TEL crystal because HCL is disconnected from salt form under solution state. If so, this evaluation makes no sense for confirmation of salt formation."
Author's answer was not adequate.
In the manuscript, the purpose of the study is still unclear, and discussion is not sufficient.
There is no evidence for the different fragmentation (What is the relation of hydrogen bonds?).
Please revise again.
Author Response
I am pleased to submit to International Journal of Pharmaceutics with entitled “Investigation of Crystallization and Salt Formation of Poorly Water-Soluble Telmisartan for Enhanced Solubility”. We have tried to address adequately the reviewer’s comments in a point-by-point manner in revised manuscript and marked in red color. Thank you very much for his/her valuable consideration.
Reviewer #1 comments | Responses |
Method: In my previous comment, "How is the composition ratio of drug and acid? If the salt is formed by method III, composition ratio of drug and salt should be calculated. And determine by analytical technique." There is still no information in the manuscript. Author answered that the composition was 1:1, but not mentioned in the method. And my question is not for preparation, what is the actual ratio of drug and HCl in "salt form"? Also, the detail for preparation method of method I-III should be described. As it is, no one can reproduce. | Thank you for your pinpointing. We have added detailed description about preparation method in the method section (Method I, II, III). In the previous version of manuscript, we didn’t mention about any filtration process for obtaining exact salt forms of TEL. As mentioned in 2.2.1.3, final salt forms of TEL were retained by 1:1 ratio with HCl. Amount of TEL and TEL/HCl required for LC/MS quantitative analysis are described. In addition, we have added the detailed information about preparation method of salt form and crystalline form of TEL. When a TEL salt form is prepared, it is theoretically estimated that the molecules are formed as non-covalent bonds with weak interactions. As a result, the amount of TEL contained in the obtained powder with TEl/HCl was quantitatively calculated by LC/MS analysis. |
MS analysis: Me previous comment "The purpose of MS analysis is uncertain. Why the fragmentation was different between TEL and salt form? Just pH condition of the sample solution? Please check same pH condition. Under solution state, if the salt dissolved completely, MS spectrum of TEL salt can be corresponding to that of TEL crystal because HCL is disconnected from salt form under solution state. If so, this evaluation makes no sense for confirmation of salt formation." Author's answer was not adequate. | Thank you for your suggestion. In order to correspond to what you pointed out, more detailed descriptions of the LC/MS analysis methods and results for TEL and TEL/HCl, with similar relative absorbances in relation to the mass spectrum have been explained. Salt formation and dissociation to the corresponding free acid or base form is the function of the pH (pHmax). In our study, appearance of the peaks at m/z 167 and m/z 391 indicated the retention of the salt form of TEL under the experimental solution state. The solubility and stability data further confirmed the stable salt form of TEL/HCL under tested conditions. |
In the manuscript, the purpose of the study is still unclear, and discussion is not sufficient. There is no evidence for the different fragmentation (What is the relation of hydrogen bonds?). Please, revise again. | Thank you for your pinpointing. We have revised the manuscript by supplementing detailed information. In case of salt formation, the non-covalent bond is connected through the hydrogen action between the drug and the salt. Structure and molecular weight of fragmented ions generated from TEL and TEL/HCl sample prepared during LC/MS analysis at the same concentration confirmed the formation of hydrogen bond interaction between the ions of TEL and HCl. |
Reviewer 2 Report
The major focus of current manuscript is to investigate solubility enhancement of telmisatan – a poorly water soluble API, via salt formation. Various characterization techniques are used for the characterization of drug stability and crystallinity. The paper is well structured, but the writing can be improved. The authors are encouraged to address the comments below.
Comments:
1. Authors are encouraged to emphasize the novelty of current work in introduction by adding related formulation literature.
2. English can be improved throughout the manuscript, e.g., Page 1 Line 29-30; Page 2 Line54-55; Page 2 Line 70-72;…
3. How do you define ‘quite stable’? Page 1 Line 33
4. ‘The pharmacokinetic behaviors and the clinical properties of TEL have been reported [14]’, please add the summary of what has been reported to complete this sentence.
5. Page 2 Line73, correct it to ‘…examined via ….’
6. Page 2 Line 74, were broken up? Please clarify.
7. Please add DSC nitrogen purging flow rate (not critical)
8. Plotting the DSC curves together is not sound to draw a conclusion. Please add melting enthalpy and onset temperature for all your DSC results in Figures 1, 4.
9. Is it expected that salt form would enhance the drug solubility?
Author Response
I am pleased to submit to International Journal of Pharmaceutics with entitled “Investigation of Crystallization and Salt Formation of Poorly Water-Soluble Telmisartan for Enhanced Solubility”. We have tried to address adequately the reviewer’s comments in a point-by-point manner in revised manuscript and marked in red color. Thank you very much for his/her valuable consideration.
Reviewer #2 comments | Responses |
1. Authors are encouraged to emphasize the novelty of current work in introduction by adding related formulation literature. | Thank you for your comment. We have added the sentence for emphasizing the novelty of current work (Line 72-83). We have further addressed the issue of stability and solubility that has not been achieved in previous studies. |
2. English can be improved throughout the manuscript, e.g., Page 1 Line 29-30; Page 2 Line54-55; Page 2 Line 70-72;… | Thank you for your comment. We have improved the English throughout the manuscript. |
3. How do you define ‘quite stable’? Page 1 Line 33 | Thank you for your comment. We have changed the term of ‘quite stable’ and modified the sentence. |
4. The pharmacokinetic behaviors and the clinical properties of TEL have been reported [14]’, please add the summary of what has been reported to complete this sentence. | Thank you for your comment. We have added the sentence to explain about the pharmacokinetic behaviors and the clinical properties of TEL (Line 62-64) |
5. Page 2 Line73, correct it to ‘…examined via ….’ | Thank you. We have corrected it to ‘examined via’ as per your comment. |
6. Page 2 Line 74, were broken up? Please clarify | Thank you for your comment. We have clarified the meaning of sentence. |
7. Please add DSC nitrogen purging flow rate (not critical) | Thank you for your comment. We have added the information about DSC nitrogen purging flow rate. Flow rate of purging gas with nitrogen was 50ml/min. |
8. Plotting the DSC curves together is not sound to draw a conclusion. Please add melting enthalpy and onset temperature for all your DSC results in Figures 1, 4. | Thank you for your comment. We have added the enthalpy and onset temperature of all DSC curves in Figure 1 and 4. |
9. Is it expected that salt form would enhance the drug solubility? | Thank you for your comment. Based on figure 8, we have proved that our TEL/HCl improve the solubility according to three bioequivalent solvents (pH 1.2, pH 6.8 buffer and distilled water). Salt formation is a common and effective method of increasing solubility and dissolution rates of acidic and basic drugs. |
Round 2
Reviewer 1 Report
The authors have corrected the manuscript accordingly in according to reviewer's suggestions. I hope this paper will contribute to the further development of pharmaceutical technologies.